# Newly Synthesized CoFe_2−x_Dy_x_O_4_ (x = 0; 0.1; 0.2; 0.4) Nanoparticles Reveal Promising Anticancer Activity against Melanoma (A375) and Breast Cancer (MCF-7) Cells

**DOI:** 10.3390/ijms242115733

**Published:** 2023-10-29

**Authors:** Slaviţa Rotunjanu, Roxana Racoviceanu, Alexandra Mioc, Andreea Milan, Roxana Negrea-Ghiulai, Marius Mioc, Narcisa Laura Marangoci, Codruţa Şoica

**Affiliations:** 1Department of Pharmacology-Pharmacotherapy, “Victor Babes” University of Medicine and Pharmacy, Eftimie Murgu Square No. 2, 300041 Timisoara, Romania; slavita.rotunjanu@umft.ro (S.R.); alexandra.mioc@umft.ro (A.M.); roxana.ghiulai@umft.ro (R.N.-G.); codrutasoica@umft.ro (C.Ş.); 2Department of Pharmaceutical Chemistry, Faculty of Pharmacy, “Victor Babes” University of Medicine and Pharmacy, Eftimie Murgu Square No. 2, 300041 Timişoara, Romania; andreea.milan@umft.ro (A.M.); marius.mioc@umft.ro (M.M.); 3Research Centre for Pharmaco-Toxicological Evaluation, “Victor Babes” University of Medicine and Pharmacy, Eftimie Murgu Square No. 2, 300041 Timisoara, Romania; 4Petru Poni Institute of Macromolecular Chemistry, 41A Aleea Gr. Ghica Vodă, 700487 Iaşi, Romania; nmarangoci@icmpp.ro

**Keywords:** cobalt ferrites, dysprosium-doped magnetic nanoparticles, cytotoxicity, cell viability, anticancer, Western blot

## Abstract

The current study focuses on the synthesis via combustion of dysprosium-doped cobalt ferrites that were subsequently physicochemically analyzed in terms of morphological and magnetic properties. Three types of doped nanoparticles were prepared containing different Dy substitutions and coated with HPGCD for higher dispersion properties and biocompatibility, and were later submitted to biological tests in order to reveal their potential anticancer utility. Experimental data obtained through FTIR, XRD, SEM and TEM confirmed the inclusion of Dy^3+^ ions in the nanoparticles’ structure. The size of the newly formed nanoparticles ranged between 20 and 50 nm revealing an inverse proportional relationship with the Dy content. Magnetic studies conducted by VSM indicated a decrease in remanent and saturation mass magnetization, respectively, in Dy-doped nanoparticles in a direct proportionality with the Dy content; the decrease was further amplified by cyclodextrin complexation. Biological assessment in the presence/absence of red light revealed a significant cytotoxic activity in melanoma (A375) and breast (MCF-7) cancer cells, while healthy keratinocytes (HaCaT) remained generally unaffected, thus revealing adequate selectivity. The investigation of the underlying cytotoxic molecular mechanism revealed an apoptotic process as indicated by nuclear fragmentation and shrinkage, as well as by Western blot analysis of caspase 9, p53 and cyclin D1 proteins. The anticancer activity for all doped Co ferrites varied was in a direct correlation to their Dy content but without being affected by the red light irradiation.

## 1. Introduction

The world of nanotechnology is evolving rapidly, touching a myriad of fields including medical sciences; among the many types of nanoformulations, magnetic nanoparticles have triggered enormous interest due to their multiple applications in various medical areas, particularly cancer therapy and diagnosis [1]. Spinel ferrites are a class of magnetic materials that can be described by the chemical formula MO·Fe_2_O_3_, where M denotes a divalent metal ion (such as Mn, Zn, Cu, Ni, etc.) that ideally occupies tetrahedral sites, while the trivalent iron ion (Fe^3+^) occupies octahedral sites [2]. The extensive investigation of these materials is driven by their potential use as therapeutic agents [3,4], contrast agents in imaging [5], gas sensors [6], semiconductors [7], high-frequency devices [8], and data storage [9,10]. Cobalt ferrite (CoO∙Fe_2_O_3_, CoFe_2_O_4_) is classified as an inverse spinel structure having the Co^2+^ ions distributed in octahedral sites, while the Fe^3+^ ions are equally distributed between tetrahedral and octahedral sites [2,11] and possesses a diverse range of properties depending on its composition, preparation method, size and shape [12].

Cobalt ferrites were scarcely exploited in biological studies due to their intrinsic toxicity achieved through degrading cobalt ions; however, their toxicity could provide cytotoxic efficacy after cellular uptake following degradation at an acidic pH which triggers the release of Co^2+^ ions [13]. Within the same study, a comparison was performed between Co^2+^ ions in the form of CoCl_2_ and Co ferrite, showing that intrinsic cobalt toxicity cannot support its use as monotherapy against tumor growth.

The anticancer activity of CoFe_2_O_4_ nanoparticles was revealed in several cancer cell lines (liver, colon, lung and brain); in leukemic cells, Co ferrite induced selective cytotoxic effects through elevated oxidative stress [14]. CoFe_2_O_4_ nanoparticles showed strong cytotoxic effects against MCF-7 breast cancer cells by reducing their viability by approximately 63%, despite inducing weak antiradical activity [12]. Moreover, the presence of Co in the ferrite structure increases its solubility in cell culture media, presumably due to changes in structure or hydrodynamic stability, or an altered overall chemical stability. The increased dissolution leads to a higher release of toxic metal ions (Co^2+^) into culture media, followed by increased cell uptake and cytotoxicity; however, a linear correlation between dissolution and cellular uptake could not be established [15].

Various element-doped ferrites were developed in order to improve the properties of ferrites by altering the structure, crystallinity and distribution of elements between tetrahedral and octahedral sites [16]; the final structural, electrical and magnetic properties of the doped Co ferrites will depend on the amount of doping element, its valency, size and site preferences. In addition, doped ferrites can be easily synthesized as single-phase materials, which is a major advantage compared to core-shell nanoparticles or composites [17]. The anticancer properties of cobalt ferrites doped with rare earth elements have been investigated in several papers; as an example, Nd^3+^-Ce^3+^-substituted CoFe_2_O_4_ exhibited anticancer effects against HCT-116 human colon cancer cells through induced apoptosis [18]. In the series of rare earth elements, Dy^3+^ ions display the highest value of magnetic moment, which strongly influences the magnetic and electrical properties of the doped ferrite [17]; Dy-doped cobalt ferrites exhibit normal spinel cubic structure [19] and a narrower particle size distribution compared to undoped ferrites [20]. Dy has already been proven effective as a doping material for the synthesis of cobalt sulfide nanoparticles, where it induced a change in their physicochemical properties and a subsequent increased efficacy for the loaded chemotherapeutic drug [21]; similar results were reported for the dysprosium-doped nickel sulfide nanoparticles [22]. Dy-doped cobalt ferrites were previously synthesized and investigated in terms of morphological and magnetic [19,20,23], and dielectric and electric [17] properties; we were able to find only one study that emphasized the antiproliferative effect of cobalt ferrites against breast cancer cells that was attributed to ROS generation [12]. To the best of our knowledge, this is the first study involving the biological assessment of Dy-doped cobalt ferrites as anticancer agents, which includes the investigation of the underlying molecular mechanisms.

Light therapy has already been in use for many years in various therapeutic procedures, with excellent outcomes, particularly in wound healing and tissue repairing due to its beneficial effects in skin regeneration through the modulation of cellular activity and collagen expression [24]; the combination of deep penetration into the skin and absorption by respiratory chain components (such as cytochrome c oxidase) enables light in the wavelength range of 600–1300 nm (red and infrared light) to promote anti-inflammatory and healing effects while being completely non-traumatic [25]. Photobiomodulation by red and near-infrared light has the ability to generate reactive oxygen species (ROS) [26] which, in high concentration, can be particularly damaging for cancer cells by triggering oxidative stress-induced cell death [14]. Austin et al. have successfully used red light therapy to inhibit melanoma proliferation and alter tumor microenvironment by increasing ROS generation and the expression of immune markers that usually indicate favorable clinical outcome [27].

The objective of the current study was to assess the impact of dysprosium-doped cobalt ferrite as an anticancer agent against melanoma (A375) and breast cancer (MCF-7) cells, while normal keratinocytes (HaCaT) were used in order to assess their potential selectivity. The Dy-doped CoFe_2_O_4_ nanoparticles were synthesized by combustion and physicochemically characterized by means of X-ray diffraction, FTIR and vibrating sample magnetometry (VSM); their morphology was investigated through scanning electron microscopy (SEM). The antiproliferative activity was assessed using the MTT cell viability assay, followed by Western blot and morphological analysis, in order to reveal the underlying molecular mechanisms of the anticancer effects in the presence/absence of red light.

## 2. Results

### 2.1. X-ray Diffraction (XRD)

As Figure 1 shows, the undoped sample (FeCo) and the doped sample (FeCo1) consist of single-phase cobalt ferrite, according to the JCPDS card 22-1086. Distinct diffraction peaks can be easily observed at specific 2θ angles of 18.28°, 30.08°, 35.43°, 37.05°, 43.05°, 53.44°, 56.97° and 62.58°, which can be attributed to the crystallographic planes (111), (220), (311), (222), (400), (422), (511) and (440), respectively, indicative of the presence of cobalt ferrite.

The presence of DyFeO_3_ was identified using JCPDS card 47-0069 with peaks at 2θ angles of 25.78°, 31.86°, 32.91°, 33.63° and 47.04°. The observed peaks in the X-ray diffraction pattern can be attributed to the crystallographic planes (111), (020), (112), (200) and (220), which suggest the presence of DyFeO_3_ formation.

In the sample FeCo2, DyFeO_3_ was detected in minimal quantities (traces), with only one small peak of diffraction being well evidenced. The diffraction patterns for cobalt ferrite in the samples FeCo, FeCo1 and FeCo2 exhibit consistent intensity levels. However, in the case of FeCo3, the diffraction patterns for cobalt ferrite illustrate a reduction in intensity, while those for DyFeO_3_ exhibit an increase in intensity.

All XRD spectra exhibit distinct, intense and narrow peaks, indicating a good level of crystallinity in the analyzed samples.

### 2.2. Fourier Transform Infrared Spectroscopy (FTIR)

The FTIR spectra (Figure 2) of all the samples under investigation exhibit a distinct absorption band at around 560 cm^−1^ and a partial adsorption band around 400 cm^−1^ which can be attributed to the metal–oxygen stretching vibration in cobalt ferrites. Furthermore, it is important to note that the doped samples exhibit a wide absorption band situated around 3450 cm^−1^, which is characteristic for the stretching vibration of –OH.

### 2.3. Investigation of Magnetic Properties

Prior to subsequent testing, the magnetic properties of the samples were assessed both for individual compounds and the respective cyclodextrin complexes in order to identify the alterations induced by the non-magnetic element; the measurements were conducted using a vibrating sample magnetometer (VSM).

It is obvious in Figure 3 that all of the hysteresis processes exhibit comparable characteristics, with variations in magnitude. The FeCo and FeCo1 samples exhibit comparable values (approximately 25 emu/g) for the remanent mass magnetization (Mr); FeCo2 shows a small decrease in Mr (24 emu/g), meanwhile, the FeCo3 sample demonstrates a significantly lower Mr value of only 15.6 emu/g. The previously mentioned pattern persists in the case of saturation mass magnetization (Ms) values, as the value of Ms decreases from 65 emu/g for FeCo to 40 emu/g for FeCo3.

The presence of cyclodextrin in the samples leads to a notable decrease in the values of Mr and Ms, respectively; as an example, Mr of 5 emu/g and Ms of 13 emu/g were measured for FeCo-γCD, while, by comparison, the sample without cyclodextrin exhibits Mr of 25 emu/g and Ms of 65 emu/g.

### 2.4. Scanning Electron Microscopy (SEM)

Figure 4 displays the scanning electron microscopy (SEM) images acquired for each of the samples, with a magnification of 10,000×.

One can notice that the morphology of the uncoated samples exhibits a resemblance to a cavernous sponge, with agglomerated particles; in the samples containing cyclodextrin, the observed pattern is indicative of the cyclodextrin presence, likely due to its significant concentration.

### 2.5. Transmitted Electron Microscopy (TEM)

The particle dimension range for the undoped and Dy-doped samples can be observed in Figure 5. It is evident that all the samples exhibit particle overlap and possess a nearly spherical morphology. Furthermore, it can be asserted that the clustering of the nanoparticles can be attributed to the magnetic characteristics exhibited by cobalt ferrite.

The FeCo particles ranged between 22.9 and 52.4 nm, while for FeCo1, the size varied between 23.1 and 40.9 nm. Similarly, FeCo2 exhibits a particle range of 20.8 to 32.3 nm, while FeCo3 varied between 17.3 and 32.5 nm.

### 2.6. The Effect of RL and γ-Cyclodextrin FeCo Complexes on Cell Viability

Treatment with FeCo-γCD, (50, 100, 250, 500 and 1000 μg/mL) had no significant effect on non-irradiated and irradiated, normal HaCaT cell line (Figure 6). However, at 50 μg/mL, we observed a slightly increased cell viability of non-irradiated (111.02 ± 3.98) and irradiated (113.2 ± 3.18) HaCaT cells vs. the control (100%). The rest of the tested compound’s cytotoxic effect followed a trend: the cytotoxic effect on HaCaT cells increased with dysprosium concentration. In detail, FeCo1-γCD exhibited a cytotoxic effect only on irradiated cells and only at the highest concentration (1000 μg/mL: 86.69 ± 2.06) vs. the control. FeCo2-γCD, only at 1000 μg/mL, inhibited cell viability of non-irradiated HaCaT cells to 89.75 ± 2.17 and to 88.66 ± 3.36 in the case of irradiated cells. Among all of the tested compounds, FeCo3-γCD exhibited the strongest cytotoxic effect, starting with 250 μg/mL, on both non-irradiated and irradiated cells, as follows: 89.90 ± 4.11, 88.21 ± 1.98 and 87.98 ± 2.59—non-irradiated cells, and 83.09 ± 3.77, 82.33 ± 3.93 and 79.97 ± 2.85—irradiated cells (Figure 6).

Cell viability of A375 cells, exposed to RL and non-exposed, significantly decreased after treatment with FeCo-γCD, FeCo1-γCD, FeCo2-γCD and FeCo_3_-γCD at 250, 500 and 1000 μg/mL vs. the control (100%). The highest inhibition occurred when the compounds were tested at 1000 μg/mL, as follows: non-irradiated—FeCo-γCD: 56.86 ± 3.93, FeCo1-γCD: 54.65 ± 4.02, FeCo2-γCD: 57.96 ± 2.59 and FeCo_3_-γCD: 42.88 ± 3.71 and irradiated—FeCo-γCD: 66.43 ± 3.11, FeCo1-γCD: 66.35 ± 2.68, FeCo2-γCD: 68.06 ± 2.88 and FeCo_3_-γCD: 59.37 ± 3.13 (Figure 7). Only FeCo1-γCD, FeCo2-γCD and FeCo_3_-γCD decreased cell viability of A375 cells at 100 μg/mL.

In MCF-7 cells, FeCo-γCD decreased cell viability only when tested at 250 μg/mL (84.20 ± 3.51/88.52 ± 3.215), 500 μg/mL (73.41 ± 5.27/80.69 ± 1.24) and 1000 μg/mL (64.89 ± 3.1/68.21 ± 3.04) vs. the control, on both non-/irradiated cells. FeCo1-γCD, FeCo2-γCD and FeCo3-γCD significantly decreased the cell viability of non-irradiated and irradiated MCF-7 cell lines at all of the tested concentrations, with the exception of FeCo2-γCD at 50 μg/mL (Figure 8).

In order to determine if the RL had any effect on cell viability, the statistical differences between the non-irradiated control and treated cells vs. the irradiated control and treated cells were analyzed using two-way ANOVA analysis followed by Bonferroni’s multiple comparisons post-test. The results showed no statistical difference between the control of non-irradiated cells vs. the control of irradiated cells, for all cell lines used. Moreover, for normal HaCaT cell lines, there were no significant differences found between treated non-irradiated and irradiated cells (Figure 6). Interestingly, and in contrast with the results obtained on HaCaT cells, the treatment of irradiated A375 cells with 500 and 1000 μg/mL FeCo2-γCD and FeCo3-γCD produced a statistically significant weaker cytotoxic effect, compared to non-irradiated and treated cells (Figure 7); this trend was not observed with MCF-7 cells (Figure 8).

### 2.7. Morphological Changes of Cell Nuclei and Cytoskeleton—Evaluation of Apoptotic Features

The effect of FeCo-γCD, FeCo1-γCD, FeCo2-γCD and FeCo_3_-γCD (1000 μg/mL) on the cellular morphology of A375 and MCF-7 was investigated using Hoechst (nuclei) and beta-actin/Alexa Fluor 488 immunofluorescent staining (Figure 9 and Figure 10). Signs of A375 apoptotic cell death were found when all of the compounds were tested (Figure 9). More precisely, the results showed that the compounds are able to induce nuclear condensation, as observed by small nuclei that appear more brightly compared to the other normal nuclei and nuclear fragmentation (apoptotic bodies). Similar results were also reported in MCF-7, where signs of apoptotic cell death appeared after the treatment with all of the tested compounds (Figure 10).

### 2.8. Western Blot Analysis of Cyclin D1, p53 and Caspase-9

In order to elucidate the cytotoxic mechanism of all 4 γ-CD FeCo complexes, caspase-9, cyclin D1 and p53 protein expression levels were determined by Western blot analysis (Figure 11).

The results obtained after Western blot analysis revealed that the compounds do not modify caspase-9, cyclinD1 and p53 protein expression levels in normal HaCaT cells. However, significant increases in caspase-9 and p53 levels were observed in A375 cells after treatment with FeCo1-γCD, FeCo2-γCD and FeCo_3_-γCD. Similar increases in caspase-9 and p53 levels were observed also when the compounds were tested on MCF-7 cell lines. The protein expression level of cyclin D1 was decreased after all compounds were applied on A375 and MCF-7 cells (Figure 11).

## 3. Discussion

Cancer is still a major challenge worldwide that, as new remedies are discovered, finds new ways to circumvent them and develops new mechanisms to endure them. Current conventional treatments that combine cancer staging with chemotherapy, radiotherapy and surgical intervention are plagued by severe side effects due to the non-specific targeting of all rapidly dividing cells and the occurrence of drug resistance after prolonged exposure [28]. In addition, most anticancer drugs exhibit poor physicochemical properties such as poor solubility and stability that trigger poor pharmacokinetic profiles, which lead to high toxicity, inefficacy and narrow biodistribution [29].

The use of nanotechnology as both diagnostic tools and treatment agents might provide the tools to overcome these challenges and has progressed immensely in the last decade with numerous types of nanoformulations being developed continuously. Cobalt ferrites belong to the class of spinel magnetic nanoparticles and display high coercivity at room temperature and moderate magnetization [30]; they have been tested as biosensors, imaging agents, therapeutic agents through induced hyperthermia as well as drug carriers [31]. CoFe_2_O_4_ nanoparticles exhibit a dose-dependent toxicity that justifies various attempts to increase their biocompatibility in order to render them useful as anticancer agents; however, their low water solubility poses a challenge in terms of their uptake by an aqueous biological environment. The covering of nanoparticles with cyclodextrin significantly increases their water solubility and dispersion due to the cyclodextrin’s film-forming ability and inclusion function [32]; in addition, cyclodextrin provides optimized biocompatibility [33]. Therefore, we chose to use hydroxypropyl-γ-cyclodextrin (HPGCD) as a coating agent with improved water solubility and inclusion capacity, compared to natural cyclodextrin which is also frequently found in pharmaceutical formulations due to its biocompatibility [34].

Conventionally, cobalt ferrites were synthesized by sol-gel and microemulsion, sol-gel auto-combustion [35], co-precipitation [36], solvothermal [37], gamma irradiation [38], microwave-assisted [39], hydrothermal [40] techniques and thermal decomposition; each method showed advantages and challenges such as the use of organic solvents and relatively toxic surfactants or the difficulty to control the crystallinity, size and shapes of the final nanoparticles [41]. The combustion method is an effective single-step, economic and easy-to-apply method which provides additional advantages such as chemically homogenous final products displaying a high degree of crystallinity and short reaction time; moreover, the high thermal gradients and fast cooling may lead to metastable phases in one step by using the intrinsic energy of the components [42]. The combustion method allows the synthesis of various nanoparticles with tailored properties via controlling the reaction’s parameters [43]. In the current study, cobalt nitrate and iron nitrate were subjected to intense heating using glycine as fuel and resulting in the formation of CoFe_2_O_4_ without significant gas release.

The XRD technique revealed the formation of cobalt ferrite and dysprosium-doped cobalt ferrite by the combustion method; the same JCPDS card 22-1086 was used in the study conducted by Lin et al. [35], who investigated the cobalt ferrite doped with Mg^2+^ using the sol-gel auto-combustion method. The study revealed the presence of identical crystallographic planes, as reported in the current study; furthermore, when using cobalt ferrite synthesized through co-precipitation [36] as an adsorbent, the same JCPDS card was employed to confirm the formation of cobalt ferrite.

In the case of the FeCo1 sample, the XRD pattern closely resembles that of the FeCo sample, indicating the successful incorporation of Dy into the structure. Furthermore, it is important to point out that the spectra of FeCo2 exhibit only a minor disparity, specifically in the form of a small diffraction peak which is characteristic for the DyFeO_3_ phase. However, the FeCo3 sample with the higher content of Dy shows more diffraction peaks characteristic for DyFeO_3_, thus suggesting the formation of a secondary phase following the saturation of cobalt ferrite with the rare element. The DyFeO_3_ compound, known as dysprosium ferrite, has been previously examined in other studies, wherein similar characteristics to those highlighted in our current study were observed, the phase being identified with JCPDS card 47-0069 and identical Miller indices [44,45].

Xavier et al. [46] conducted an investigation on CoFe_2−x_Dy_x_O_4_ using the sol-gel method, followed by a two-step sinterization process at 500 °C. For x = 0.5; 1; 1.5, they obtained a single phase, suggesting the successful incorporation of Dy in the spinel structure. The authors concluded that in the case of x = 0.2, a secondary phase was formed, and due to the low intensity of the secondary phase peak, the majority of Dy^3+^ occupies the octahedral site of the spinel and only small amounts of Dy lead to DyFeO_3_ formation. In the present study, comparable outcomes were acquired with respect to the formation of a singular phase for the composition x = 0.1 (FeCo1), while a minor peak intensity of a secondary phase was observed for the composition x = 0.2 (FeCo2). Similar behaviors were reported for other rare earth elements; Zhao et al. [47] employed the emulsion method to synthesize CoFe_2−x_Nd_x_O_4_ (x = 0; 0.1; 0.15; 0.2), which was subjected to sinterization at various temperatures. The successful incorporation of Nd^3+^ ions into the spinel structure of cobalt ferrite was achieved, even at a value of x = 0.2, without the occurrence of any secondary phase formation. The synthesis of CoGd_x_Fe_2−x_O_4_ (x = 0; 0.25; 0.5) was conducted by Javed et al. [48], who noticed the formation of GdFeO_3_ as a secondary phase, with small peak intensities for x = 0.25 and with intense peaks for x = 0.50. In our case, for Dy^3+^-doped ferrite, we obtained only a small peak for the secondary phase for x = 0.2; however, for x = 0.4, more diffraction peaks characteristic of DyFeO_3_ were present on the XRD spectra, in addition to those associated with cobalt ferrite.

In the present study, both cobalt ferrite and doped cobalt ferrite exhibit comparable IR spectral absorption bands at approximately 565 cm^−1^ and 400 cm^−1^. Similar results were reported by Khizar et al. [49], whose cobalt ferrite sample exhibited a prominent absorption band at 565 cm^−1^ which was attributed to the metal–oxygen vibration. Furthermore, the spectral region ranging from 400 to 565 cm^−1^ was specifically assigned to the vibrational behavior exhibited by the oxygen–metal cation complexes within the octahedral and tetrahedral sites of the spinel structure. Other studies also reported the same allocations of the absorption bands from this domain [47,50,51]. The small differences within the 400–565 cm^−1^ range between the doped and undoped samples can be assigned to the incorporation of Dy^3+^ ions into the spinel structure, as elucidated in Yadav’s investigation [23]; Ansari et al. [52] made similar observations in their study on the effects of dysprosium and copper substitution in cobalt ferrite. The Dy-doped samples also exhibited a band located around 3450 cm^−1^ that can be attributed to the –OH stretching vibration [53,54].

Magnetic measurements identified decreased Mr and Ms values for the doped samples in direct proportion to their increase in Dy content; the same pattern was reported by Javed et al. [48], who investigated cobalt ferrite doped with Gd^3+^ ions in terms of magnetic properties and the impact of gadolinium on these properties. The study indicated a proportional decrease in both remanent magnetization (Mr) and saturation magnetization (Ms) as the Gd^3+^ content increased. Additionally, an increase in Gd content resulted in the formation of a secondary phase, GdFeO_3_, in a similar manner to DyFeO_3_ in the current study. Comparable findings were achieved by Xavier et al. for Dy-doped cobalt ferrite [46], where an increase in dysprosium content to x = 0.25 resulted in a reduction of approximately 40% in the Ms values; in our study, a notable decrease in values was observed when the dysprosium content was x = 0.40. In the context of neodymium-doped cobalt ferrite, the Ms value diminishes as the concentration of Nd^3+^ increases, while maintaining a consistent sample size [47]. The substitution of iron by a rare earth element in the octahedral site of the spinel structure is attributed to the different ionic radius and magnetic moment of the rare earth element; this substitution results in a reduction in the Fe^3+^–Fe^3+^ interaction, consequently leading to a decrease in both magnetic moment and Ms values [46,47,48].

SEM and TEM studies were employed in order to assess the morphology of newly synthesized nanoparticles, revealing sponge-like structures for both doped and undoped samples; similar findings were reported for CoAl_x_Fe_2−x_O_4_ obtained by the sol-gel combustion method [55], Co_x_Fe_3−x_O_4_ synthesized by the solution combustion method [56] and CoGd_x_Fe_2−x_O_4_ prepared by the sol-gel auto-combustion method [57], indicating that the use of similar synthetic approaches leads to comparable morphological features. The introduction of Dy into the structure of cobalt ferrites did not alter the morphology of the resulting nanoparticles, as also highlighted by Yadav et al. [23], but did have an effect on the particle size. The dimensions of the undoped sample ranged from 20 to 50 nm, while the doped samples exhibited a range of 17 to 40 nm; also, the particle size decreased as the Dy content increased. Both types of samples displayed spherical shapes. Aziz and Azhdar [20] reported comparable findings, suggesting that the undoped sample exhibited greater particle dimensions compared to Dy-doped nanoparticles, an additional influencing factor being the grinding type employed. Similar results were obtained by Ansari et al. [52] in the case of Dy^3+^- and Cu-substituted cobalt ferrite, by Abbas et al. [55] in the case of Al^3+^-doped cobalt ferrite and by Dixit et al. [58] for Ce- and Gd-doped nickel ferrite nanoparticles. Nanoparticles ranging in diameter between 10 and 100 nm were indicated as highly suitable for anticancer therapy, with smaller particles being able to leak from normal vessels and damage healthy cells, while larger particles are subjected to fast clearance by phagocytes [59]. Tang et al. investigated the optimal size anticancer nanoparticles should display in order to be effective and reported that the 50 nm diameter induced the strongest inhibitory effect against primary and metastatic tumors [60]. In addition, nanoparticles ranging between 20 and 50 nm exhibited a slow clearance rate that prolonged their circulation time; moreover, the disintegration of larger nanoparticles into fragments of 20–50 nm in size significantly improved their penetration into deep tumor tissue [61]. Our results, in terms of nanoparticle size, fit within the size range mentioned above, thus ensuring optimal anticancer activity.

One main objective of the current study was to assess the cytotoxicity of all types of ferrites, with or without Dy doping, against both normal and cancer cells in order to establish an optimal proportion of iron and Dy ions in ferrites with anticancer potential. All nanoparticles were formulated as complexes with HPGCD that provided a dispersant effect, thus enabling the preparation of stable suspensions which could be applied more easily on cell cultures. We used healthy HaCaT keratinocytes as the target to assess the cytotoxic effects of doped and undoped Co ferrites against normal cells, given the fact that selectivity stands as an important parameter for any anticancer agent that may reach further development. For the undoped Co ferrite, no significant effect was recorded regardless of the applied concentration with or without red light irradiation; moreover, slight cell proliferation occurred for the lowest concentration, unlike when higher concentrations were used, presumably due to the mechanical effect of ferrite suspension on keratinocytes. These results are in line with previous studies that reported the lack of cytotoxic effects against normal kidney cells using the same concentration range [62]; cytotoxic effects appeared, in fact, at significantly higher concentrations (up to 4 mg/mL) in a dose-dependent manner when CoFe_2_O_4_ nanoparticles were applied on HaCaT cells [63]. A similar lack of toxicity against HaCaT cells was noticed by Zhang et al., who prepared Zn-Cr-doped Co ferrites and reported a slight increase in cell viability when used in small concentrations (up to 200 μg/mL) through a yet unidentified mechanism [64]. In our study, the Dy-doped CoFe_2_O_4_ nanoparticles displayed a clear direct correlation between the cytotoxic effects and Dy proportion in the nanoparticles’ structure; however, the cytotoxic effects were noticed only for high concentrations, thus revealing adequate biocompatibility for all types of nanoparticles. Irradiation with red light exerted only a slight influence on cell viability percentages; generally, irradiation led to a very subtle stronger cytotoxicity.

In A375 human melanoma cells, a significant cytotoxic effect was recorded in a dose-dependent manner, the highest percentage of cell death being achieved when the highest concentration (1000 μg/mL) of all types of nanoparticles was applied; the cytotoxic effect of tested samples was comparable to the one recorded for 5-fluoruracil, conventionally used as anticancer chemotherapy. When different proportions of Dy were compared, the data showed a direct correlation between the strength of the cytotoxic effect and the percentage of the doping element in the CoFe_2_O_4_ structure. The positive influence of Dy presence on the potency of the antiproliferative activity is clearly reflected by the occurrence of such activity at lower concentrations (starting from 100 μg/mL) only for doped Co ferrites. To the best of our knowledge, this is the first study focused on the cytotoxic activity of Dy-doped Co ferrites, although their structure and physicochemical characterization has been investigated [20]; however, Dy was previously reported as a doping element for cobalt sulfide nanoparticles that revealed intrinsic anticancer activity against MCF-7 breast cancer cells but also the potential to serve as a drug carrier [21]. In terms of red light irradiation, experiments showed that a stronger cytotoxic effect occurred when cells were not irradiated, which contradicts previously published data by Austin et al., who reported the antiproliferative activity of red light phototherapy against melanoma cells through oxidative stress and apoptosis. Conversely, Ayuk et al. attributed red light photobiomodulation with the ability to increase cell proliferation and viability in fibroblasts, in direct correlation with the level of cell stress [65]. It is therefore possible that in our study, the cytotoxic effect of RL in melanoma cells reported by Austin et al. was countered by its proliferative effect developed as a result of cell stress induced by Dy-doped Co ferrites.

In MCF-7 breast cancer cells, a similar significant dose-dependent cytotoxic effect was noticed for all types of doped ferrites, with a direct correlation with the Dy proportion; the undoped ferrite sample also induced cytotoxic effects in a dose-dependent manner but only starting with the 250 μg/mL concentration. Therefore, we can safely state that Dy doping has a positive influence on the cytotoxic activity displayed by ferrite nanoparticles in tested cells. So far, cobalt ferrites were tested as drug carriers for the delivery of docetaxel in MCF-7 cancer cells, where the authors reported effective cellular uptake [66]; however, the study also showed an intrinsic cytotoxicity displayed by the bare Co ferrite nanoparticles even for lower concentrations than those used in the current study. By contrast, Yeganeh et al. synthesized CoFe_2_O_4_ nanoparticles as drug carriers for letrozole and reported almost no sign of cytotoxicity of the bare nanoparticles in MCF-7 cells; however, in this case, magnetic nanoparticles were coated with methionine, a physiological amino acid, in a core-shell structure that may hamper the intrinsic toxicity of the core ferrites [67]. In another study, Co ferrites were able to decrease cell viability in MCF-7 cells but only when used in concentrations as high as 2 mg/mL [12]. In MCF-7 cells, the influence of red light irradiation did not exhibit statistical significance.

Apoptosis is the process of natural, organized, highly regulated cell death identified as a molecular mechanism for numerous anticancer agents, since delayed or inhibited apoptosis is a marker for cancer and autoimmunity [68]. We investigated the potential apoptosis induction by CoFe_2_O_4_ nanoparticles by means of Hoechst staining (nuclei) and beta-actin assessment (cytoskeleton), which revealed nuclear condensation and fragmentation together with a disrupted cytoskeleton in both types of treated cancer cells, A375 and MCF-7. During apoptosis, dedicated caspases are activated and start targeting the nuclear skeleton, leading to nuclear fragmentation, which has long been identified as an apoptotic marker; caspases also target the microtubule cytoskeleton, which will be dismantled relatively early in the process, leading to cellular shrinkage and fragmentation [69]. Our data clearly show the occurrence of an apoptotic process identifiable in both nuclei and cell cytoskeleton; in fact, anticancer agents with antimicrotubule activity (taxane, Vinca alkaloids) have long been introduced very successfully in the anticancer arsenal. Apoptosis was indicated as a molecular mechanism for the antiproliferative effect of Ce–Nd co-substituted nanospinel cobalt ferrites in HCT116 colon cancer cells [18], as well as for nickel-doped Co ferrites in MCF-7 breast cancer cells [70] and Ce-Dy co-activated Mn–Zn nanospinel ferrites [71]. Horev-Azaria et al. established that Co ferrites toxicity in various cell lines is dependent on cell type and occurs through the generation of ROS, which presumably leads to apoptosis [72]; similar results were reported by Kanagesan et al. in 4T1 breast cancer cells [73]. To the best of our knowledge, this is the first study regarding the molecular mechanisms underlying the cytotoxic properties of Dy-doped Co ferrites and the experiments revealed a somewhat expected result—apoptosis induction—which was identified as well for other types of doped cobalt ferrites.

In order to further clarify the intimate mechanisms involved in the apoptotic cell death induced by the spine nanoparticles, we performed Western blot analysis of cyclin D1, p53 and caspase-9 expression levels; in both types of cells, increased levels of caspase-9 and p53 protein were recorded for all four types of cobalt ferrites, with the highest levels corresponding to the largest Dy substitution of iron ions in doped ferrites. The undoped ferrite sample induced the lowest apoptotic effect; in normal HaCaT cells, no change in the levels of the two markers was detected. Also, one can notice a slightly higher susceptibility of A375 cells to Dy-doped Co ferrites compared to MCF-7 cells. Caspase-9 is a key player in the regulation of intrinsic apoptosis, its activation failure leading to severe diseases including cancer and reflecting the potential development of drug resistance to chemotherapy [74]. p53 protein exhibits tumor suppression as its main function; therefore, its activation stands as a viable anticancer strategy [75]. As previously mentioned, we were not able to find published data regarding the molecular mechanisms of apoptosis for Dy-doped ferrites; however, Ni–Zn ferrites were found to induce selective toxicity in cancer cells by activating caspase-9 and up-regulating p53 protein expression, leading to the collapse of the mitochondrial membrane potential [76]. Similarly, activation and increased expression of caspase-9 and the cell cycle checkpoint protein p53 were reported in lung cancer cells treated with nickel ferrite nanoparticles [77]; the authors also revealed that the apoptotic effect of their ferrites depended on the cell type [78], which corresponds to our own observations, the activation of caspase-9 being clearly more pronounced in A375 cells. Apoptosis induction is validated as well through the decreased expression of cyclin D1, which is an essential regulator in cell cycle, its overexpression being linked to cancer development and progression [79]; collectively, experimental data indicate that Dy-doped Co ferrites act as efficient and selective anticancer agents in both types of cancer cells through mitochondrial apoptosis induction, as revealed by caspase-9, p53 and cyclin D1 regulated expressions.

All data considered, the newly developed metallic nanoparticles provide the potential to be used as anticancer treatment against both melanoma and breast cancer by using appropriate administration routes; in melanoma, the nanoparticles could be embedded in a topical delivery formulation such as hydrogels that possess unique properties in both improving anticancer effects and minimizing side effects [80]. In breast cancer, the nanoparticles can be administered via the parenteral route, such as injection in the tail vein [81]; however, future in vivo studies are needed in order to clarify such aspects. Also, our results clearly revealed the intrinsic cytotoxicity of the newly synthesized doped ferrites, which could therefore be used as such in anticancer treatments; however, previous studies reported the potential of nanoferrites to be used as delivery systems for other anticancer agents following surface functionalization [82]. Future studies will focus on the development of delivery platforms based on the currently described nanoparticles, taking into consideration the experimental data reported herein that may serve as reference in the assessment of anticancer efficacy.

## 4. Materials and Methods

### 4.1. Chemicals

The raw materials employed in the experiment included cobalt nitrate hexahydrate (Co(NO_3_)_2_·6H_2_O), iron nitrate nanohydrate (Fe(NO_3_)_3_·9H_2_O), glycine (C_2_H_5_NO_2_), dysprosium chloride hexahydrate (DyCl_3_·6H_2_O), (2-Hydroxypropyl)-γ-cyclodextrin and ethanol absolute. The cobalt nitrate, dysprosium chloride and (2-Hydroxypropyl)-γ-cyclodextrin were obtained from Sigma-Aldrich (St. Louis, Missouri, United States of America), while the iron nitrate, glycine and ethanol absolute were acquired from Merck (Rahway, NJ, USA).

### 4.2. Synthesis by Combustion of Cobalt Ferrite—FeCo

The experimental procedure involved the use of 0.02 moles of cobalt nitrate, 0.04 moles of iron nitrate and 0.09 moles of glycine, calculated in order to obtain 0.02 moles of CoFe_2_O_4_. All the raw materials were placed in a Berzelius glass and heated continuously at 60 °C until complete dissolution in their crystallization water. The obtained brown solution was subsequently transferred into a porcelain capsule and positioned on a heating mantle preheated at approximately 350 °C. Following the complete water evaporation, the mixture underwent a boiling phase and then self-ignition; the propagation of the combustion front was observed to occur rapidly, with a duration of approximately 9 s. During the reaction, observable yellow flames were present, signifying the attainment of exceedingly high temperatures. Furthermore, the reaction progressed without any substantial gas release. Upon the conclusion of the reaction, the emergence of a black yet dense and slightly crumbly powder was observed.

### 4.3. Synthesis of Cobalt Ferrite Doped with Dysprosium

The methodology described for the undoped sample (Section 2.2) was replicated while incorporating dysprosium chloride in three distinct proportions (see Table 1); the same cobalt and glycine content was used. White gases were released during the combustion reaction. When compared to the undoped sample, there was a decrease in both flame intensity and temperature. Additionally, the duration of the reaction increased by approximately 4–5 s. Following the completion of the reaction, the entire precursor mixture underwent a conversion process, yielding a voluminous black powder with a spongy texture.

### 4.4. Cyclodextrin Complex Formation

To evaluate the efficacy of ferrites as anticancer agents, the ferrite particles were initially incorporated into cyclodextrin. A molar ratio of 1:1 between ferrite and hydroxypropyl gamma cyclodextrin was employed. The homogenization of the mixture was achieved by using 3 mL of solvent composed of 2.1 mL absolute ethanol and 0.9 mL distilled water. The mixture was manually blended for 30 min and then dried in the oven at 70 °C; subsequently, each sample was manually grounded.

For the biological experiment, a suspension was prepared, which was sonicated for 2 h in a UP200S ultrasonic homogenizer (Hielscher Ultrasonics GmbH, Teltow, Germany). The sonication process was conducted with 0.8 cycles and 50% amplitude. At the end of the process, no discernible particles were noticed at the bottom of the flask.

### 4.5. Characterization Methods

The investigation of the sample’s phase composition was conducted using X-ray diffraction (XRD) analysis. The evaluation was performed using a Rigaku Ultima IV instrument (Tokyo, Japan), which operated at a voltage of 40 kV and a current of 40 mA. The XRD pattern was obtained by utilizing CuKα radiation. The FTIR spectra were obtained using KBr pellets on a Shimadzu IR Affinity-1S spectrophotometer (Shimadzu Scientific Instruments Inc., Columbia, MD, USA) within the range of 400–4000 cm^−1^, with a resolution of 4 cm^−1^. Magnetic measurements were performed at room temperature on a LakeShore 8607 vibrating sample magnetometer (VSM, Shore Cryotronics, Westerville, OH, USA), at a magnetic field range of 30 to −30 KOe. Prior to each test, the samples subjected to VSM analysis were demagnetized in alternating field. Scanning electron microscopy (SEM) was performed on a Verios G4 UC scanning electron microscope (Thermo Scientific, Czech Republic). The samples were coated prior to examination with 6 nm platinum using a Leica EM ACE200 Sputter coater in order to increase electrical conductivity and reduce charge buildup, which can occur during exposure to the electron beam. SEM investigations were performed in high vacuum mode using a detector for high-resolution images (Through Lens Detector, TLD) at an accelerating voltage of 10 kV, the magnification being indicated on the micrographs. Morphological characterization of the samples was carried out in high contrast mode at 120 kV acceleration voltage on a Hitachi High-Tech HT7700 (Hitachi High-Technologies Corporation, Tokyo, Japan). The instrument is equipped with a STEM module, an energy dispersive X-ray (EDX) detector that allows elemental analysis and selected area electron diffraction (SAED) apertures which could be used to collect diffraction patterns. The samples were prepared by drop casting from their water suspension on 300 mesh carbon-coated copper grids (Ted Pella) and vacuum-dried at room temperature for 24 h.

### 4.6. Biological Assessment

#### 4.6.1. Cell Culture

HaCaT immortalized human keratinocytes were purchased from CLS Cell Lines ServiceGmbH (Eppelheim, Germany), and A375 human malignant melanoma (ATCC^®^ CRL-1619^TM^) and MCF-7 human breast adenocarcinoma (ATCC^®^ HTB-22™) cell lines were purchased from ATTC American Type Culture Collection (Łomianki, Poland). The cells were cultured in appropriate media: HaCaT and A375 cell lines in Dulbecco’s Modified Eagle Medium (DMEM) and MCF-7 cell lines in Eagle’s Minimum Essential Medium (EMEM). All media used were supplemented with 10% FBS and 1% penicillin/streptomycin mixture (10,000 IU/mL) (Thermo Fisher Scientific, Boston, MA, USA). The cells were grown under standard conditions in a humidified incubator with a 5% CO_2_ atmosphere at 37 °C.

#### 4.6.2. Cell Viability Assay

Cell viability was measured by the MTT (3-(4,5-Dimethylthiazol-2-yl)-2,5-Diphenyltetrazolium Bromide) reduction assay (Roche, Mannheim, Germany). Briefly, the cells were cultured in 96-well plates (1 × 10^4^ cells/well). After reaching appropriate confluency, the cells were treated with four γ-cyclodextrin FeCo complexes: FeCo-γCD, FeCo1-γCD, FeCo2-γCD and FeCo3-γCD (50, 100, 250, 500 and 1000 μg/mL) and with γ-cyclodextrin (1000 μg/mL); 5-FU (2, 5, 10, 25, 50 μg/mL) served as a positive control. Following a 48-hour treatment period, the plates were incubated with 10 μL/well of MTT reagent for 3 h at 37 °C, and then with 100 μL/well of MTT buffer solution for 30 min at room temperature and in the dark. Absorbance was measured at 570 nm using a xMark™ Microplate Spectrophotometer (Bio-Rad Laboratories, Inc., Hercules, CA, USA).

#### 4.6.3. Red Light (RL) Experimental Protocol

HaCaT, A375 and MCF-7 cells seeded in 96-well plates (1 × 10^4^ cells/well) were treated with FeCo1-γCD, FeCo2-γCD and FeCo3-γCD (50, 100, 250, 500 and 1000 μg/mL) and incubated for 2 h in the dark. After the incubation, the cells were irradiated for 25 min with 40 J/cm^2^ of light (red and infrared light, λ: 640 ± 30 nm and 880 ± 30 nm) emitted by a Medolight Z4L device (Zepter Group, Bioptron AG, Switzerland).

#### 4.6.4. Detection of Apoptotic Morphological Changes

HaCaT, A375 and MCF-7 cells treated for 48 h with the highest concentration of FeCo1-γCD, FeCo2-γCD and FeCo_3_-γCD were washed with PBS, fixed with methanol for 15 min and permeabilized (Triton X 0.01% in PBS) for another 15 min. The cells were then blocked for 30 min (3% BSA) at room temperature and stained with beta-actin monoclonal antibody (1:2000, MA1-140, Thermo Fisher Scientific, Inc., Waltham, MA, USA) for 1 h at room temperature. Next, the cells were treated with Alexa Fluor 488 Goat-anti-Mouse Secondary Antibody (1:1000, Thermo Fisher Scientific, Inc., Waltham, MA, USA) for 30 min in the dark, at room temperature. Finally, the cells’ nuclei were stained with Hoechst 33,342 nuclear staining assay (1:2000 in PBS, Thermo Fisher Scientific, Boston, MA, USA) for 10 min at room temperature and in the dark. The changes in nuclear morphology were observed using the inverted fluorescent microscope Optika IM-3FL4 (Optika, Ponteranica, Italy) equipped with CellSens Entry Software v 3.0 (Olympus, Tokyo, Japan) and DP28-CU camera (Olympus, Tokyo, Japan).

#### 4.6.5. Western Blot

The cells were cultured in a 6-well plate (1x10^6^ cells/well) and treated with the highest concentration (1000 μg/mL) of FeCo, FeCo_1_, FeCo_2_ and FeCo_3_ for 48 h at 37 °C. Cells were then scrapped and lyzed with RIPA buffer (Thermo Fisher Scientific, Inc., Waltham, MA, USA). Equal amounts of proteins (20 μg, Pierce™ Rapid Gold BCA Protein Assay Kit, Thermo Fisher Scientific, Inc., Waltham, MA, USA) were loaded in each lane of a Novex^®^ NuPAGE^®^ 4–12% Bis-Tris gel (NP0321BOX, Thermo Fisher Scientific, Inc., Waltham, MA, USA). The proteins were then separated (Mini Gel Tank-A25977, Thermo Fisher Scientific, Inc., Waltham, MA, USA) and transferred onto a nitro-cellulose membrane by iBlot^®^ 2 Dry Blotting System (IB23001, Thermo Fisher Scien-tific, Inc., Waltham, MA, USA). The membranes were blocked with 5% skimmed milk and then probed with 0.5 µg/mL cyclin D1 antibody (33–3500, Thermo Fisher Scientific, Inc., Waltham, MA, USA), 1,5 µg/mL p53 monoclonal antibody (MA5-12453, Thermo Fisher Scientific, Inc., Waltham, MA, USA) and 1 µg/mL caspase-9 polyclonal antibody (PA5-16358, Thermo Fisher Scientific, Inc., Waltham, MA, USA) in blocking buffer at 4 °C overnight. Beta-tubulin mouse monoclonal antibody 1:1000 dilution (32–2600, Thermo Fisher Scientific, Boston, MA, USA) was used as a loading control. Goat anti-Mouse IgG (H+L) Superclonal™ Recombinant Secondary Antibody, HRP (1:2000), and Goat anti-Rabbit IgG (H+L) Superclonal™ Secondary Antibody, HRP conjugate (1:4000, Thermo Fisher Scientific, Inc., Waltham, MA, USA), were used as a secondary antibody. Pierce™ ECL Western blotting substrate and ChemiDoc MP imaging system (170–8280, BioRad, Hercules, CA, USA) were used for the chemiluminescent detection (Thermo Fisher Scientific, Inc., Waltham, MA, USA). The acquisition and analysis of the images were performed using the Image Lab software v6.1 (BioRad, Hercules, CA, USA).

#### 4.6.6. Statistical Analysis

GraphPad Prism version 6.0.0 (GraphPad Software, San Diego, CA, USA) was used for the statistical analysis. The differences between groups were considered statistically significant if *p <* 0.05, as follows: * *p <* 0.05, ** *p <* 0.01, and *** *p <* 0.001.

## 5. Conclusions

While the physical properties of rare earth-substituted nanosized cobalt ferrites were thoroughly investigated, their biological effects particularly in terms of anticancer efficacy remain generally poorly known. The current study focused on the straightforward synthesis of Dy-doped CoFe_2_O_4_ nanospinels via combustion, followed by their physicochemical assessment. Three types of doped nanoparticles were prepared containing different Dy substitutions and coated with HPGCD for higher dispersion properties and biocompatibility, and were later submitted to biological tests in order to reveal their potential anticancer utility. Cell viability assay indicated strong antiproliferative activity for all doped Co ferrites in a direct correlation to their Dy content but without being affected by the red light irradiation. Mitochondrial apoptosis was identified as the underlying mechanism of cell death induced by spinel nanoparticles, as revealed by increased expressions of caspase-9 and p53 tumor suppression protein and decreased levels of cyclin D1. Further studies are warranted in order to assess the in vivo anticancer potential of such magnetic nanoparticles which could also be used as drug carriers.

## Figures and Tables

**Figure 1 ijms-24-15733-f001:**
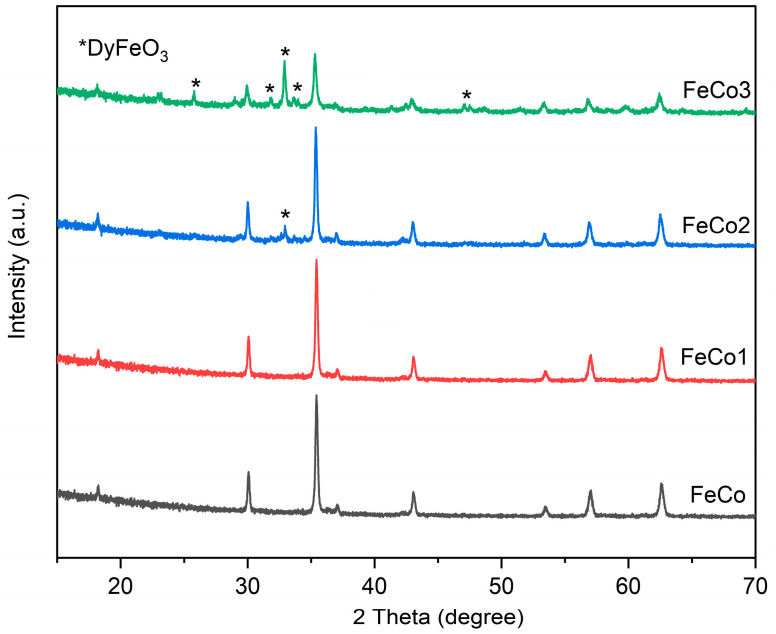
XRD patterns of samples FeCo, FeCo1, FeCo2 and FeCo3.

**Figure 2 ijms-24-15733-f002:**
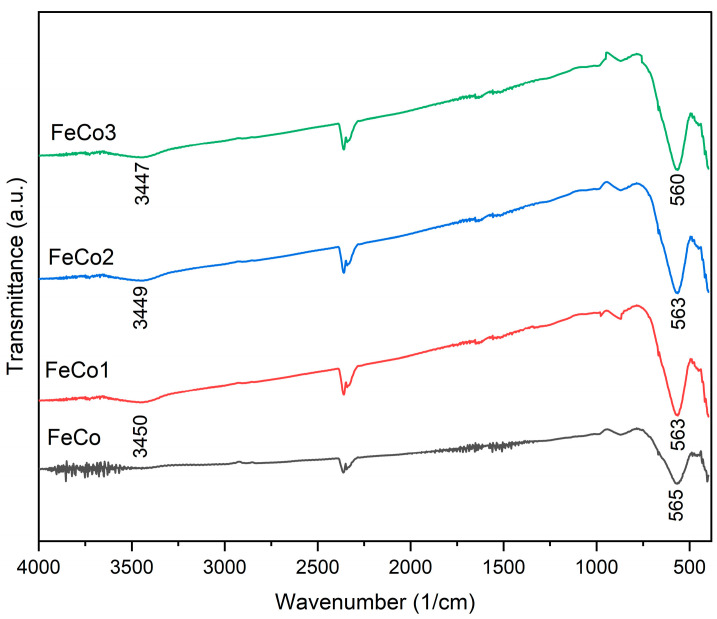
FTIR spectra of samples FeCo, FeCo1, FeCo2 and FeCo3.

**Figure 3 ijms-24-15733-f003:**
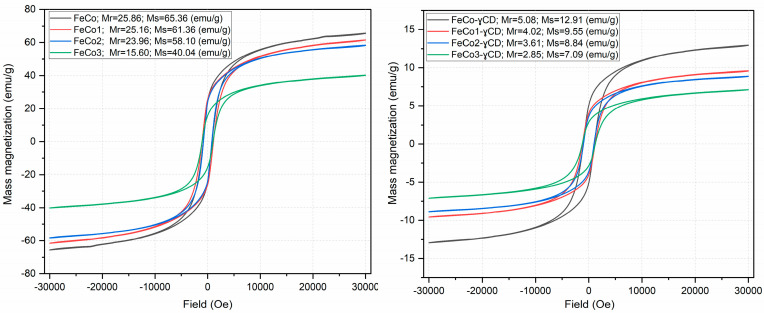
Magnetic hysteresis of samples.

**Figure 4 ijms-24-15733-f004:**
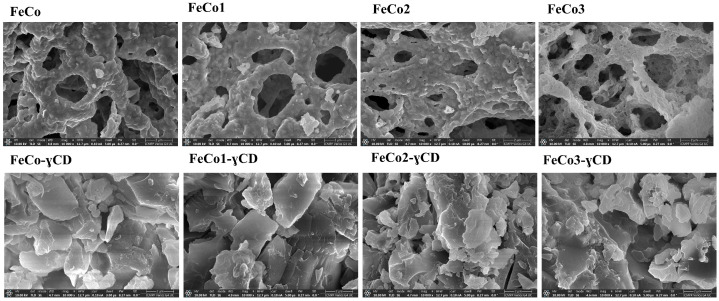
SEM images of the samples; scale bar 2 µm.

**Figure 5 ijms-24-15733-f005:**
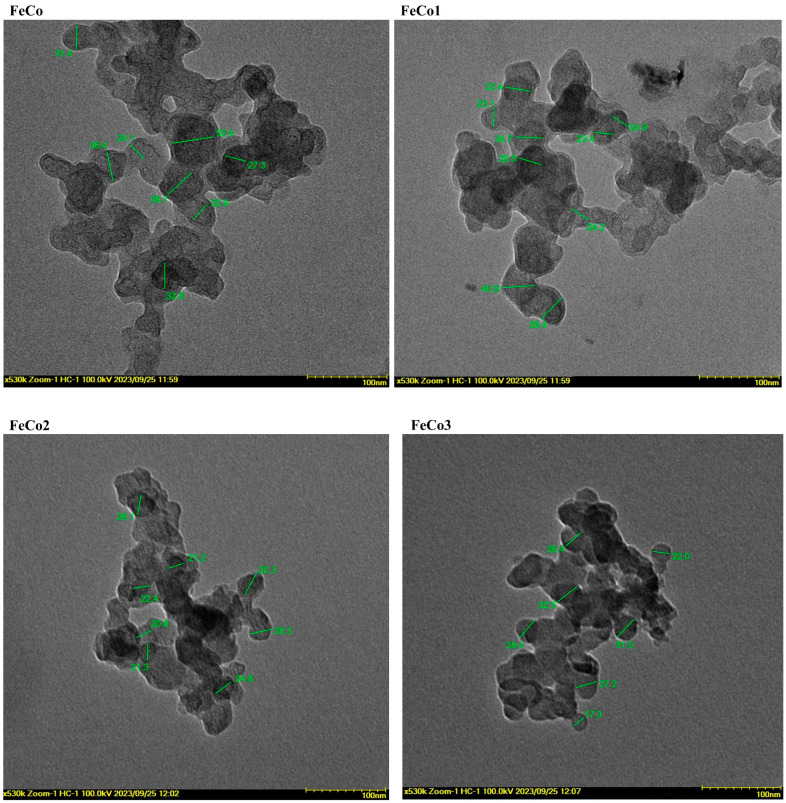
TEM images of the samples FeCo, FeCo1, FeCo2 and FeCo3; scale bar 100 nm.

**Figure 6 ijms-24-15733-f006:**
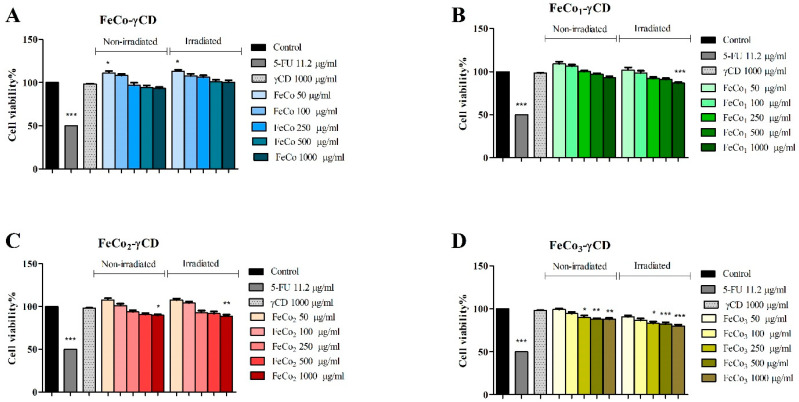
Cell viability of HaCaT cells, non-irradiated and irradiated with PDT cells, treated for 48 h with 50, 100, 250, 500 and 1000 μg/mL FeCo-γCD (**A**), FeCo1-γCD (**B**), FeCo2-γCD (**C**) and FeCo_3_-γCD (**D**). The results are expressed as cell viability percentage (%) normalized to control (100%). The data represent the mean values ± SD of three independent experiments performed in triplicate. The statistical difference vs. control was determined using two-way ANOVA analysis followed by Bonferroni’s multiple comparisons post-test (* *p* < 0.05, ** *p <* 0.005 and *** *p <* 0.0001).

**Figure 7 ijms-24-15733-f007:**
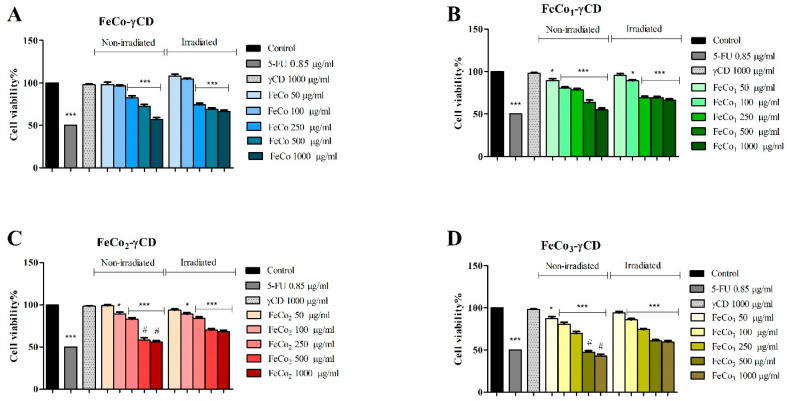
Cell viability of A375 cells, non-irradiated and irradiated with RL cells, treated for 48 h with 50, 100, 250, 500 and 1000 μg/mL FeCo-γCD (**A**), FeCo1-γCD (**B**), FeCo2-γCD (**C**) and FeCo_3_-γCD (**D**). The results are expressed as cell viability percentage (%) normalized to control (100%). The data represent the mean values ± SD of three independent experiments performed in triplicate. The statistical difference vs. control was determined using two-way ANOVA analysis followed by Bonferroni’s multiple comparisons post-test (**p <* 0.05 and *** *p <* 0.0001). The statistical difference between the non-irradiated and irradiated cells is marked with # (*p <* 0.05).

**Figure 8 ijms-24-15733-f008:**
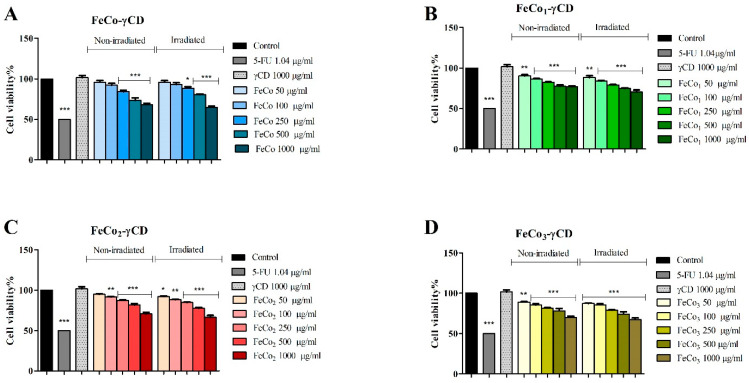
Cell viability of MCF-7 cells, non-irradiated and irradiated with PDT cells, treated for 48 h with 50, 100, 250, 500 and 1000 μg/mL FeCo-γCD (**A**), FeCo1-γCD (**B**), FeCo2-γCD (**C**) and FeCo_3_-γCD (**D**). The results are expressed as cell viability percentage (%) normalized to control (100%). The data represent the mean values ± SD of three independent experiments performed in triplicate. The statistical difference vs. control was determined using two-way ANOVA analysis followed by Bonferroni’s multiple comparisons post-test (* *p <* 0.05, ** *p <* 0.005 and *** *p <* 0.0001).

**Figure 9 ijms-24-15733-f009:**
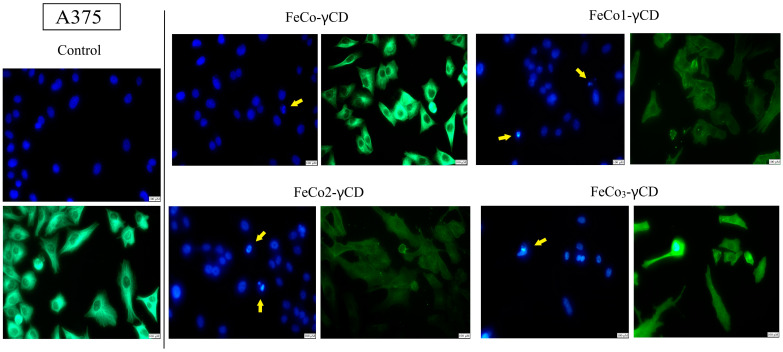
Morphological assessment of A375 nuclei (blue staining) and cytoskeleton (beta-actin—green staining) after treatment with 1000 μg/mL FeCo-γCD, FeCo1-γCD, FeCo2-γCD and FeCo_3_-γCD. The yellow arrows represent signs of apoptotic cell death: small and brightly stained nuclei (nuclear condensation) and fragmentation. The scale bar was 100 μm.

**Figure 10 ijms-24-15733-f010:**
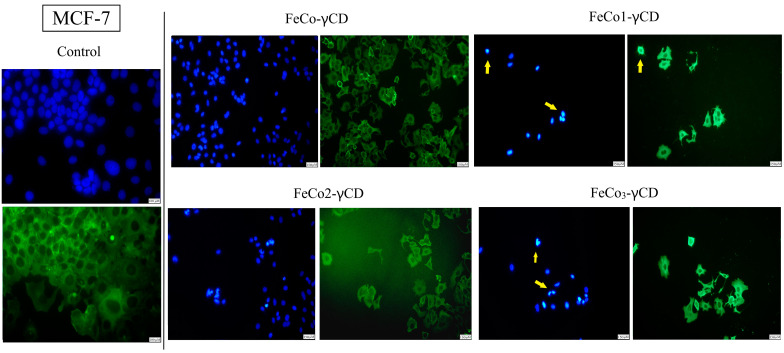
Morphological assessment of MCF-7 nuclei (blue staining) and cytoskeleton (beta-actin—green staining) after treatment with 1000 μg/mL FeCo-γCD, FeCo1-γCD, FeCo2-γCD and FeCo_3_-γCD. The yellow arrows represent signs of apoptotic cell death: small and brightly stained nuclei (nuclear condensation) and fragmentation. The scale bar was 100 (control) and 150 μm.

**Figure 11 ijms-24-15733-f011:**
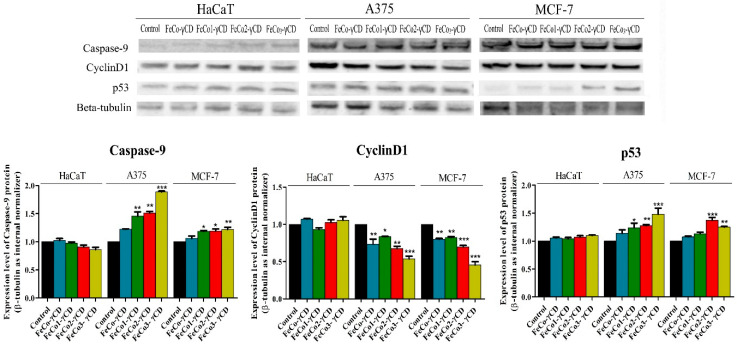
Determination of caspase-9, cyclin D1 and p53protein expression in HaCaT, A375 and MCF-7 cells after treatment with 1000 μg/mL FeCo-γCD, FeCo1-γCD, FeCo2-γCD and FeCo_3_-γCD. The results were normalized against the beta-tubulin loading control and the control group. The statistical differences vs. control were determined using one-way ANOVA analysis followed by Tukey’s multiple comparisons post-test (* *p <* 0.05, ** *p <* 0.01 and *** *p <* 0.001).

**Table 1 ijms-24-15733-t001:** Doped sample composition.

Sample	DyCl_3_·6H_2_O (Moles)	Fe(NO_3_)_3_·9H_2_O (Moles)
FeCo1	0.002	0.038
FeCo2	0.004	0.036
FeCo3	0.008	0.032

## Data Availability

Not applicable.

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
