# Peer review of "Newly Synthesized CoFe2−xDyxO4 (x = 0; 0.1; 0.2; 0.4) Nanoparticles Reveal Promising Anticancer Activity against Melanoma (A375) and Breast Cancer (MCF-7) Cells"

_ijms, 2023, doi:10.3390/ijms242115733_

Round 1

Reviewer 1 Report

Comments and Suggestions for Authors

The current experimental research manuscript on the development of novel metallic nanoparticles for cancer treatment is quite interesting, including many relevant studies, with the performed analysis being thorough. Hence, I only ask that the following changes are made before acceptance for publication:

- The safety of the developed nanoparticles should be addressed, since they are made of metallic components;

- The limitations of current cancer therapy should be mentioned thoroughly, in order to better support the need for the development of new therapies;

- Obtained particle size should be discussed more thoroughly: what are the ideal sizes for the intended application?;

- Future studies should be further discussed, including the possible administration route and formulation type, for example topical administration for melanoma? And inserting the nanoparticles within a hydrogel? Or other viable options;

- Considering a possible future administration, would the developed nanoparticles serve better as therapeutics, or as drug carriers? Or both? Additionally, should they be used as adjuvant therapies for current treatments, or as main therapy?

- The novelty of the developed nanoparticles should be better supported.

Author Response

Dear reviewer,

Thank you very much for taking your time to review our manuscript entitled “Newly synthesized CoFe2-xDyxO4 (x=0; 0.1; 0.2; 0.4) nanoparticles reveal promising anticancer activity against melanoma (A375) and breast cancer (MCF-7) cells”. We really appreciate all your comments and suggestions. Those comments are all valuable and very helpful for revising and improving our paper, as well as the important guiding significance to our researches. The comments were studied carefully, and corrections have been made accordingly in the revised manuscript.

The current experimental research manuscript on the development of novel metallic nanoparticles for cancer treatment is quite interesting, including many relevant studies, with the performed analysis being thorough. Hence, I only ask that the following changes are made before acceptance for publication:

- The safety of the developed nanoparticles should be addressed, since they are made of metallic components;

Answer: We tested all the metallic nanoparticles on healthy HaCaT keratinocytes specifically to investigate the level of their toxicity since selectivity is an essential issue in the anticancer activity. According to our study, the ferrite nanoparticles do not alter in a significant manner the viability of normal cells, results that were discussed in detail in the Discussions section. For the Dy-doped nanoparticles a very low toxicity was noticed but only for the highest tested concentration. Therefore, we might expect the same results in vivo but the actual in vivo testing is needed in order to formulate such a hypothesis.

- The limitations of current cancer therapy should be mentioned thoroughly, in order to better support the need for the development of new therapies;

Answer: The text was improved with more details concerning the challenges of the current standard of care (lines 296-305). Two new references were added to better support our statements.

- Obtained particle size should be discussed more thoroughly: what are the ideal sizes for the intended application?;

Answer: The manuscript was corrected according to suggestions, see lines 415-426.

- Future studies should be further discussed, including the possible administration route and formulation type, for example topical administration for melanoma? And inserting the nanoparticles within a hydrogel? Or other viable options;

Answer: The necessary information was added as suggested, see lines 538-545.

- Considering a possible future administration, would the developed nanoparticles serve better as therapeutics, or as drug carriers? Or both? Additionally, should they be used as adjuvant therapies for current treatments, or as main therapy?

Answer: We added the necessary information in the manuscript, see lines 545-552.

- The novelty of the developed nanoparticles should be better supported.

Answer: The necessary information was added in the manuscript, see lines 85-91.

Reviewer 2 Report

Comments and Suggestions for Authors

In this work, the authors report a new CoFe2-xDyxO4 nanoparticle family to achieve a promising anticancer activity against melanoma (A375) and breast cancer (MCF-7) cells. Overall, this manuscript can be accepted after well addressing the following important issues.

1.      In Figure 1, the XRD patterns of FeCo and FeCo1 are the same.

2.      Necessary references should be added to support the analysis of FTIR in Figure 2. For example, the position of OH- stretching vibration in the FTIR plot can refer to this work (DOI: 10.1063/5.0083059).

3.      There are no scale bars in Figure 5 and the words in this figure are unclear.

4.      Besides the morphology and structure of the samples, electronic structural properties of the synthesized materials should be compared and measured via XPS for Co, Fe, and O elements, which are very important information. The fitting and analysis can refer to this work (DOI: 10.1039/D1TA10652J).     

Comments on the Quality of English Language

Minor editing of English language required

Author Response

Dear reviewer,

Thank you very much for taking your time to review our manuscript entitled “Newly synthesized CoFe2-xDyxO4 (x=0; 0.1; 0.2; 0.4) nanoparticles reveal promising anticancer activity against melanoma (A375) and breast cancer (MCF-7) cells”. We really appreciate all your comments and suggestions. Those comments are all valuable and very helpful for revising and improving our paper, as well as the important guiding significance to our researches. The comments were studied carefully, and corrections have been made accordingly in the revised manuscript.

In this work, the authors report a new CoFe2-xDyxO4 nanoparticle family to achieve a promising anticancer activity against melanoma (A375) and breast cancer (MCF-7) cells. Overall, this manuscript can be accepted after well addressing the following important issues.

  1. In Figure 1, the XRD patterns of FeCo and FeCo1 are the same.

Answer: Due to the successful incorporation of Dy into the cobalt ferrite structure, the FeCo and FeCo1 samples exhibit similar XRD patterns without any diffractions peaks that would indicate the formation of a secondary phase.

  1. Necessary references should be added to support the analysis of FTIR in Figure 2. For example, the position of OH- stretching vibration in the FTIR plot can refer to this work (DOI: 10.1063/5.0083059).

Answer: The reference suggested by the reviewer has been added to the manuscript, see lines 381-382.

  1. There are no scale bars in Figure 5 and the words in this figure are unclear.

Answer: Figure 5 was modified and a visible scale bar was added.

  1. Besides the morphology and structure of the samples, electronic structural properties of the synthesized materials should be compared and measured via XPS for Co, Fe, and O elements, which are very important information. The fitting and analysis can refer to this work (DOI: 10.1039/D1TA10652J).     

Answer: We totally agree with the reviewer that XPS analysis brings important information regarding the electronic state of elements present in the samples; however, and we apologize for this, our infrastructure do not allow us to conduct this particular analysis at the moment. Also, the literature we found in the biomedical field on such metallic nanoparticles do not usually include the analysis of electronic structural properties of the synthesized materials as they do not correlate directly to the biological effect which was our main objective. We confirmed the formation of cobalt ferrites through a variety of analysis (XRD, FTIR, VSM, SEM and TEM) that also provided information in terms of size, morphology and magnetic properties which are the parameters with direct implications in biological effects as revealed by the literature in the field (which was cited in the manuscript). We kindly ask the reviewer to take these data into consideration since the article is meant as an interdisciplinary approach with the final goal of discovering new anticancer agents with high biological effects.